# 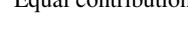 GoLLIE : Annotation Guidelines improve Zero-Shot Information-Extraction

**Oscar Sainz***, **Iker García-Ferrero***
**Rodrigo Agerri**, **Oier Lopez de Lacalle**, **German Rigau**, **Eneko Agirre**
HiTZ Basque Center for Language Technology - Ixa NLP Group
University of the Basque Country (UPV/EHU)
{oscar.sainz, iker.garciaf}@ehu.eus

## Abstract

Large Language Models (LLMs) combined with instruction tuning have made significant progress when generalizing to unseen tasks. However, they have been less successful in Information Extraction (IE), lagging behind task-specific models. Typically, IE tasks are characterized by complex annotation guidelines that describe the task and give examples to humans. Previous attempts to leverage such information have failed, even with the largest models, as they are not able to follow the guidelines out of the box. In this paper, we propose GoLLIE (**G**uideline-f**o**llowing **L**arge **L**anguage Model for **IE**), a model able to improve zero-shot results on unseen IE tasks by virtue of being fine-tuned to comply with annotation guidelines. Comprehensive evaluation empirically demonstrates that GoLLIE is able to generalize to and follow unseen guidelines, outperforming previous attempts at zero-shot information extraction. The ablation study shows that detailed guidelines are key for good results. Code, data, and models are publicly available: https://github.com/hitz-zentroa/GoLLIE.

## 1 Introduction

The task of Information Extraction (IE) is highly challenging. This challenge is evident in the detailed guidelines, which feature granular definitions and numerous exceptions, that human annotators must follow to perform the task. The performance of current SoTA models heavily depends on the quantity of human-annotated data, as the model learns the guidelines from these examples. However, this performance significantly decreases when tested in new annotation schema (Liu et al., 2021a). The common practice in IE to achieve good results is to manually annotate each new domain and schema from scratch, as almost no transfer exists across application domains. Unfortunately, this is unfeasible, both, in terms of financial cost and human effort.

Recent advancements in Large Language Models (LLM) (Min et al., 2023) have enabled the development of models capable of generalizing to unseen tasks. Thus, current zero-shot IE systems

---

*Equal contribution

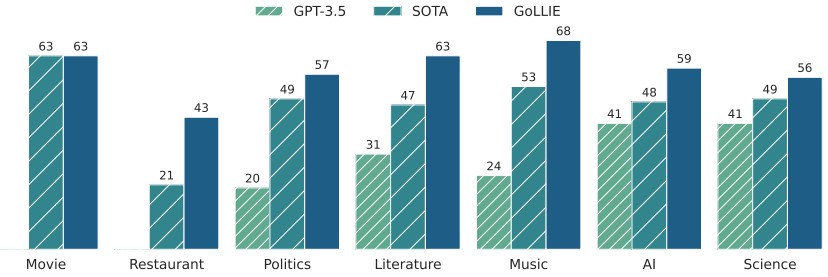

Figure 1: Out of domain zero-shot NER results. GPT results are not available for all domains.

leverage the knowledge encoded in LLMs to annotate new examples (Sainz et al., 2022a; Wang et al., 2023a). As a by-product of the pre-training process, models possess now a strong representation of what a person or an organization is. Therefore, they can be prompted to extract mentions of those categories from a text. However, this has a clear limitation: not every annotation schema* defines "person" (or any other label) in the same way. For example, ACE05 (Walker et al., 2006) annotates pronouns as persons, while, CoNLL03 (Tjong Kim Sang & De Meulder, 2003) does not. IE tasks require more information than just label names, they require annotation guidelines.

Current LLMs have been trained to follow instructions, but they fail to follow annotation guidelines out of the box. For instance, Figure 1 shows results on domain-specific zero-shot Named Entity Recognition. The results of gpt-3.5-turbo when prompted with guidelines (Zhang et al., 2023a) are low, around 20 F1 score on Music or Politics domains. Building a system that enables high-performance zero-shot information extraction, reducing the dependence on costly human annotations, remains an open challenge.

In this work, we present 🦙 GoLLIE (**G**uideline-f**o**llowing **L**arge **L**anguage Model for **IE**), an LLM fine-tuned to learn how to attend to the guidelines on a small set of well known IE tasks. Comprehensive zero-shot evaluation empirically demonstrates that GoLLIE outperforms the SoTA (Wang et al., 2023a) in zero-shot information extraction (see Figure 1).

## 2 RELATED WORK

Large Language Models (LLMs) have made significant advancements toward the development of systems that can generalize to unseen tasks (Min et al., 2023). Radford et al. (2019) trained LLMs using a vast amount of internet data, finding that pre-trained models given natural language task descriptions can perform tasks such as question answering, machine translation, or summarizing without explicit supervision. Building on this discovery, instruction tuning, often referred to as multitask fine-tuning, has emerged as the leading method to achieve generalization to unseen tasks. This process involves pre-training a model on a massive amount of unlabeled data and subsequently fine-tuning it on a diverse collection of tasks (Wang et al., 2022; Chung et al., 2022) phrased as text-to-text problems (Raffel et al., 2020). A natural language instruction or prompt is given to the model to identify the task it should solve (Schick & Schütze, 2021; Scao & Rush, 2021). Research has demonstrated that increasing the parameter count of the language model (Brown et al., 2020), coupled with improvements in the size and quality of the instruction tuning dataset, results in enhanced generalization capabilities (Chen et al., 2023; Zhang et al., 2022; Chowdhery et al., 2022; Muennighoff et al., 2023; Touvron et al., 2023a;b). LLMs have displayed impressive zero-shot generalization capabilities in various challenging tasks, including coding Wang & Komatsuzaki (2021); Black et al. (2022); Rozière et al. (2023), common sense reasoning Touvron et al. (2023a), and medical applications Singhal et al. (2023), among others.

In the field of Information Extraction (IE), recent shared tasks (Fetahu et al., 2023) have shown that encoder-only language models such as XLM-RoBERTa (Conneau et al., 2020) and mDEBERTA (He et al., 2023) remain the most effective models. Attempts to utilize LLMs and natural language instructions for IE have been less successful (Tan et al., 2023; Zhou et al., 2023; Zhang et al., 2023a), as their performance lags behind that of encoder-only models. Before the billion parameters LLMs, indirectly supervised methods improve zero-shot IE by utilizing the knowledge learned from tasks like Textual Entailment (Sainz et al., 2021; 2022a;b) and Question Answering (Levy et al., 2017). Obeidat et al. (2019) propose an entity typing method that encodes label descriptions from Wikipedia as embeddings using an LSTM, which is then used to score the inputs. Methods that leveraged external knowledge were also successful on fine-grained zero-shot NER (Chen et al., 2021). Lu et al. (2022a) introduced a unified text-to-structure generation that can model different IE tasks universally. Lou et al. (2023) proposed converting IE tasks to a semantic matching problem, allowing their method to generalize to new domains and label ontologies not seen during training. Wang et al. (2023a) framed IE tasks as natural language descriptive instructions and trained an LLM across a diverse range of IE tasks. In evaluations on tasks with unseen label ontologies, their model outperformed other instruction-tuning methods.

---

*We define schema as the set of labels and their definitions.

```
Schema definition          # The following lines describe the task definition
                           @dataclass
                           class ProgrammingLanguage(Entity):
Guidelines are introduced      """Refers to a programming language used in the development of AI
as docstrings                  applications and research. Annotate the name of the programming
                               language, such as Java and Python."""

Representative candidates
are introduced as comments     span: str   # Such as: "Java", "R", "CLIPS", "Python", "C + +"

                           @dataclass
                           class Metric(Entity):
Labels are defined as          """Refers to evaluation metrics used to assess the performance of AI
python classes                 models and algorithms. Annotate specific metrics like F1-score."""

                               span: str   # Such as: "mean squared error", "DCG", …

Input text                 # This is the text to analyze
                           text = "Here , accuracy is measured by error rate , which is defined as..."

Output annotations         # The annotation instances that take place in the text above are listed here
                           result = [
Annotations are                Metric(span="accuracy"),
represented as instances       Metric(span="error rate"),
                           ]
```

Figure 2: Example of the input and output of the model.

Most instruction tuning attempts for IE share a limitation: they only consider label names in the prompts (e.g., *"List all the Persons"*). This poses two major challenges. Firstly, not all datasets share the same definition for labels like *Person* (some exclude fictional characters or pronouns). Secondly, a label name alone doesn't sufficiently describe complex or less common labels. While there have been attempts to prompt LLMs using guidelines (Zhang et al., 2023a), strong prior knowledge of LLMs regarding task labels (Blevins et al., 2023) deter the model from adhering to those guidelines.

## 3 APPROACH

Different from previous approaches, 🐻 GoLLIE forces the model to attend to the details in the guidelines, performing robustly on schemas not seen during training. On this section we deep dive into the details of our approach, describing how the input and output was represented and the regularization techniques used to force the model to attend to the guidelines.

### 3.1 INPUT-OUTPUT REPRESENTATION

We have adopted a Python code-based representation (Wang et al., 2023b; Li et al., 2023) for both the input and output of the model. This approach not only offers a clear and human-readable structure but also addresses several challenges typically associated with natural language instructions. It enables the representation of any information extraction task under a unified format. The inputs can be automatically standardized using Python code formatters such as Black. The output is well-structured and parsing it is trivial. Furthermore, most current LLMs incorporate code in their pre-training datasets, indicating that these models are already familiar with this representation.

Figure 2 shows the three main parts of the format: schema definition, input text, and output annotations. **Schema definition** forms the initial segment of the input. This section contains information about the labels that are represented as Python classes; guidelines, articulated as docstrings; and representative annotation candidates presented in the form of code comments. The number of class definitions corresponds to the number of labels in the dataset. Classes are flexible and vary for each task. For example, classes for a NER dataset merely require an attribute to specify the text span that corresponds to the class. On the other side, more complex tasks such as Event Argument Extraction (EAE) or Slot Filling (SF) demand more class attributes to categorize the task, such as a list of participants in an event (refer to examples in Appendix A). **Input text** is the second part of the input. The input text is represented as a string variable in Python. **Output annotations** is the part generated by the model. The model starts generating after `result =`. The annotations are represented as a list of instances of the classes defined on the schema definition part. Parsing the output is straightforward; executing the generated code in Python yields a list containing the result. This ease of parsing the output stands as a significant advantage of our model. A further detailed analysis of the efficiency of this approach is available in Appendix E.

```
@dataclass                                      @dataclass
class VulnerabilityPatch(Event):                class VulnerabilityPatch(Event):
                                                    """A VulnerabilityPatch Event happens when a software company addresses a
                                                    known vulnerability by releasing or describing an appropriate update."""

    mention: str                                    mention: str
    cve: List[str]                                  """The text span that triggers the event.
    issues_addressed: List[str]                     Such as: patch, fixed, addresses, implemented, released
    supported_platform: List[str]                   """
    vulnerability: List[str]                        cve: List[str] # The vulnerability identifier
    vulnerable_system: List[str]                    issues_addressed: List[str] # What did the patch fix
    releaser: List[str]                             supported_platform: List[str] # The platforms that support the patch
    patch: List[str]                                vulnerability: List[str] # The vulnerability
    patch_number: List[str]                         vulnerable_system: List[str] # The affected systems
    system_version: List[str]                       releaser: List[str] # The entity releasing the patch
    time: List[str]                                 patch: List[str] # What was the patch about
                                                    patch_number: List[str] # Number or name of the patch
                                                    system_version: List[str] # The version of the vulnerable system
                                                    time: List[str] # When was the patch implemented, the date
```

Figure 3: Example of the input representation. (left) An example of an event definition w/o guidelines information. (right) The same example but with guideline information as Python comments.

## 3.2 GUIDELINES ENHANCED REPRESENTATION

The main contribution of this work is the use of the guidelines as part of the inference process to improve the zero-shot generalization. An example of a class definition with and without guidelines is shown in Figure 3. Different datasets usually define guidelines in many different ways: some provide a complex definition of a label with several exceptions and special treatments and others just give a few representative candidates of the fillers of the label. To normalize the input format, we included the label definitions as class docstrings and the candidates as a comment for the principal argument (which is usually *mention* or *span*). Complex tasks such as EAE or SF require additional definitions for the arguments or slots, to that end, we included small definitions as comments on each class argument. In this paper, we will refer to the model without guidelines as Baseline and the model with guidelines as 🦙 GoLLIE.

## 3.3 TRAINING REGULARIZATION

We want to ensure that the model follows the guidelines and does not just learn to identify specific datasets and perform correctly on them. To do this, we introduce various kinds of noise during training. This stops the model from recognizing particular datasets, recalling specific labels, or attending only to the label names rather than learning to follow the actual description for each label in the guidelines.

We applied the following regularizations. **Class order shuffling**, for each example, the order of the input classes is randomly shuffled. This makes it more difficult for the model to memorize entire task definitions. **Class dropout**, we delete some of the input classes randomly. By eliminating few classes from both the input and output, we force the model to learn to only output instances of classes defined in the input. This not only encourages the model to focus on the schema definition but also minimizes the occurrence of hallucinations during inference. **Guideline paraphrasing**, we generate variations of the label definitions to prevent the model from easily memorizing them. We also think this will make the method more robust to different variations on the definition. **Representative candidate sampling**, similar to what we do with the paraphrases, for each input we sample 5 different candidates from a fixed pool of 10 per class. **Class name masking** involves substituting the label class names (e.g., PERSON) with placeholders, such as LABEL_1. This prevents the model from exploiting the label names during training and forces it to attend and understand the guidelines.

## 4 EXPERIMENTAL SETUP

### 4.1 DATA

Evaluating zero-shot capabilities requires dividing the data into training and evaluation datasets. However, many benchmarks for Information Extraction are based on the same domain or share part of their schema. To ensure that the zero-shot evaluation is not affected by similar data, we have divided our set of benchmarks based on the domain of the data (a related topic is data contamination,

Table 1: Datasets used on the experiments. The table shows the domain, tasks and whether are use for training, evaluation or both.

| Dataset | Domain | NER | RE | EE | EAE | SF | Training | Evaluation |
|---------|--------|-----|----|----|-----|----|----------|------------|
| ACE05 (Walker et al., 2006) | News | ✓ | ✓ | ✓ | ✓ | | ✓ | ✓ |
| BC5CDR (Wei et al., 2016) | Biomedical | ✓ | | | | | ✓ | ✓ |
| CoNLL 2003 (Tjong Kim Sang & De Meulder, 2003) | News | ✓ | | | | | ✓ | ✓ |
| DIANN (Fabregat et al., 2018) | Biomedical | ✓ | | | | | ✓ | ✓ |
| NCBIDisease (İslamaj Doğan & Lu, 2012) | Biomedical | ✓ | | | | | ✓ | ✓ |
| Ontonotes 5 (Pradhan et al., 2013) | News | ✓ | | | | | ✓ | ✓ |
| RAMS (Ebner et al., 2020) | News | | | | ✓ | | ✓ | ✓ |
| TACRED (Zhang et al., 2017) | News | | | | | ✓ | ✓ | ✓ |
| WNUT 2017 (Derczynski et al., 2017) | News | ✓ | | | | | ✓ | ✓ |
| BroadTwitter (Derczynski et al., 2016) | Twitter | ✓ | | | | | | ✓ |
| CASIE (Satyapanich et al., 2020) | Cybercrime | | | ✓ | ✓ | | | ✓ |
| CrossNER (Liu et al., 2021b) | *Many* | ✓ | | | | | | ✓ |
| E3C (Magnini et al., 2021) | Biomedical | ✓ | | | | | | ✓ |
| FabNER (Kumar & Starly, 2022) | Science | ✓ | | | | | | ✓ |
| HarveyNER (Chen et al., 2022) | Twitter | ✓ | | | | | | ✓ |
| MIT Movie (Liu et al., 2013) | Queries | ✓ | | | | | | ✓ |
| MIT Restaurants (Liu et al., 2013) | Queries | ✓ | | | | | | ✓ |
| MultiNERD (Tedeschi & Navigli, 2022) | Wikipedia | ✓ | | | | | | ✓ |
| WikiEvents(Li et al., 2021) | Wikipedia | ✓ | | ✓ | ✓ | | | ✓ |

which we discuss in Appendix G). For training we kept mostly datasets from **News and Biomedical** domains, for evaluation instead, we used datasets from **diverse domains**. This approach helps to avoid introducing any noise into the evaluation process. Among the evaluation datasets we included CrossNER (Liu et al., 2021b), a dataset that is split into many domains, for simplicity, we will call each domain as a separate dataset: AI, Literature, Music, Politics, and Science. Also, we will refer to MIT Movie and MIT Restaurant as Movie and Restaurant. Table 1 contains the information about the data used in the experiments.

We have trained the model to perform 5 different tasks: Named Entity Recognition (NER), Relation Extraction (RE), Event Extraction (EE), Event Argument Extraction (EAE), and Slot Filling (SF). However, we only evaluated the model on the three main tasks of interest: NER, EE, and EAE. The other two tasks are added to the training data to add diversity and improve the flexibility of the model.

A few modifications have been made to two datasets to improve the quality of the model. First, the training data of Ontonotes 5 was reduced drastically as it was automatically annotated. Second, the TACRED dataset was converted from RE to SF to increase the complexity of the task. These modifications make our system not comparable with the state of the art on those tasks. However, our focus of interest is on the zero-shot evaluation and, therefore, the benefits (see Appendix A) are more interesting than adding 2 more comparable points on the supervised setup. In the CASIE dataset, we detected that the annotated event spans are inconsistent. The models typically annotate a sub-string rather than the entire span. Therefore, we evaluate all the models based on the predicted event categories, without considering the exact text span. For arguments, we use partial matching.

We use the guidelines released by the authors of each dataset (More details in Appendix F). When such guidelines are not publicly available, we ask human experts to create them, based on the annotations from the development split. The representative candidates are extracted from the guidelines when available, otherwise, the candidates are sampled from the the train split based on word frequency or manually curated based on the guidelines. Paraphrases are automatically generated using Vicuna 33B v1.3 (Zheng et al., 2023).

## 4.2 LANGUAGE MODELS AND BASELINES

**Backbone LLMs:** GoLLIE is a fine-tuned version of Code-LLaMA Rozière et al. (2023). Other backbone LLMs, such as LLaMA (Touvron et al., 2023a), LLaMA-2 Touvron et al. (2023b) or Falcon Penedo et al. (2023) were considered during the development, however, as our approach uses code to represent the input and output, Code-LLaMA model worked better on the preliminary experiments. In order to perform fair comparisons the baseline developed in this paper is based on Code-LLaMA as well. All the development of this paper was done with the 7B parameter version of Code-LLama, but, for a scaling analysis, we also trained the 13B and 34B parameter models.

Table 2: Supervised evaluation results. "*" indicates that results are not directly comparable.

| Dataset | SoTA | Baseline | 🦙 | 🦙 13B | 🦙 34B |
|---------|------|----------|-----|--------|--------|
| $ACE05_{NER}$ | (Wang et al., 2023a) 86.6 | $89.1_{\pm0.2}$ | $88.1_{\pm0.6}$ | $89.4_{\pm0.2}$ | $\mathbf{89.6}_{\pm0.1}$ |
| $ACE05_{RE}$ | (Lu et al., 2022b) 66.1 | $63.8_{\pm0.6}$ | $63.6_{\pm1.8}$ | $67.5_{\pm0.5}$ | $\mathbf{70.1}_{\pm1.5}$ |
| $ACE05_{EE}$ | (Lu et al., 2022b) $\mathbf{73.4}$ | $71.7_{\pm0.2}$ | $72.2_{\pm0.8}$ | $70.9_{\pm1.6}$ | $71.9_{\pm1.1}$ |
| $ACE05_{EAE}$ | (Lu et al., 2022b) *54.8 | $65.9_{\pm0.7}$ | $66.0_{\pm0.8}$ | $67.8_{\pm0.9}$ | $\mathbf{68.6}_{\pm1.2}$ |
| BC5CDR | (Zhang et al., 2023b) $\mathbf{91.9}$ | $87.5_{\pm0.2}$ | $87.5_{\pm0.2}$ | $87.9_{\pm0.1}$ | $88.4_{\pm0.2}$ |
| CoNLL 2003 | (Lu et al., 2022b) 93.0 | $92.9_{\pm0.1}$ | $92.8_{\pm0.3}$ | $93.0_{\pm0.2}$ | $\mathbf{93.1}_{\pm0.1}$ |
| DIANN | (Zabala et al., 2018) 74.8 | $80.3_{\pm0.7}$ | $79.4_{\pm1.1}$ | $82.6_{\pm1.3}$ | $\mathbf{84.1}_{\pm1.1}$ |
| NCBIDisease | (Wang et al., 2023a) $\mathbf{90.2}$ | $86.2_{\pm0.1}$ | $85.4_{\pm0.3}$ | $86.5_{\pm0.8}$ | $85.8_{\pm0.2}$ |
| Ontonotes 5 | - | $83.4_{\pm0.2}$ | $83.4_{\pm0.2}$ | $84.0_{\pm0.2}$ | $\mathbf{84.6}_{\pm0.4}$ |
| RAMS | (Li et al., 2021) 48.6 | $48.9_{\pm0.4}$ | $48.7_{\pm0.7}$ | $49.6_{\pm0.1}$ | $\mathbf{51.2}_{\pm0.3}$ |
| TACRED | - | $56.6_{\pm0.2}$ | $57.1_{\pm0.9}$ | $56.7_{\pm0.5}$ | $\mathbf{58.7}_{\pm0.2}$ |
| WNUT 2017 | (Wang et al., 2021) $\mathbf{60.2}$ | $53.7_{\pm0.7}$ | $52.0_{\pm0.6}$ | $50.5_{\pm0.9}$ | $54.3_{\pm0.4}$ |
| Average | | $73.3_{\pm0.1}$ | $73.0_{\pm0.3}$ | $73.9_{\pm0.3}$ | $\mathbf{75.0}_{\pm0.3}$ |

**Training setup:**    To train the models we use QLoRA (Hu et al., 2022; Dettmers et al., 2023). LoRA freezes the pre-trained model weights and injects trainable rank decomposition matrices into linear layers of the Transformer architecture. In a preliminary experiment, this setup outperformed fine-tuning the entire model on the zero-shot tasks, while training much faster (more details in Appendix D.4). We applied the LoRA to all linear transformer block layers as recommended by Dettmers et al. (2023). The models were trained for 3 epochs with an effective batch size of 32 and a learning rate of 3e-4 with a cosine scheduler. Our training infrastructure was 2 NVIDIA's A100 with 80gb each. More details about the training are given in the Appendix D.

**Comparable systems:**    Our main point of comparison is Instruct-UIE (Wang et al., 2023a) as it is the approach closest to our system, but does not use guidelines. Another system considered for comparison is PromptNER (Zhang et al., 2023a), which proposes to prompt GPT-3.5 and T5 with definitions using Chain-of-Though in order to perform few-shot NER. Different from us, they did not fine-tune the model to attend to the guidelines. For fair comparison, we only considered the zero-shot results reported in the paper. In addition, other SoTA systems are added for comparison when results from Instruct-UIE and PromptNER are not available. Given that our system is designed for the zero-shot scenario, the supervised experiments are intended to verify that our system does not degrade its performance. Thus we selected, for the supervised scenario, those systems among SoTA that share the most comparable setting with us.

## 5    RESULTS

### 5.1    SUPERVISED EVALUATION

The results on the supervised datasets are shown in Table 2. Comparing GoLLIE with the baseline, they both obtain very similar results, with an absolute difference of 0.3 F1 points on average. This is expected, as the baseline model implicitly learns the guidelines for annotating the datasets based on the data distribution during fine-tuning. In addition, despite the noise introduced to GoLLIE fine-tuning in order to generalize from guidelines, the performance is close to that of the baseline.

Compared to other systems our model achieves similar results in general. Focusing on the two datasets where our model under-performs significantly, WNUT and NCBIDisease, we find that task-specific techniques are still needed. For instance, Wang et al. (2021) uses external knowledge to detect emergent and rare entities. In the NCBIDisisease dataset, models pre-trained on Biomedical domain corpora achieve the best results (Kocaman & Talby, 2021). (Wang et al., 2023a) leverages Flan-T5, which has great proficiency on Biomedical domain tasks (Singhal et al., 2022). These improvements, however, are complementary to our proposal.

### 5.2    ZERO-SHOT EVALUATION

The results on the zero-shot are shown in Table 3. Overall, compared to the baseline, **the results are improved significantly when using guidelines** on almost every dataset, with an absolute difference

Table 3: Zero-shot evaluation results. "*" indicates results obtained using the original code.

| Dataset | SoTA | Baseline | 🦙 | 🦙 13B | 🦙 34B |
|---|---|---|---|---|---|
| BroadTwitter | - | $39.0_{\pm0.6}$ | $49.5_{\pm0.8}$ | $\mathbf{51.4}_{\pm1.8}$ | $50.3_{\pm2.1}$ |
| $CASIE_{EE}$ | - | $33.9_{\pm6.5}$ | $59.3_{\pm2.3}$ | $62.2_{\pm0.9}$ | $\mathbf{65.5}_{\pm1.8}$ |
| $CASIE_{EAE}$ | - | $47.9_{\pm1.4}$ | $50.0_{\pm1.1}$ | $52.6_{\pm0.2}$ | $\mathbf{55.2}_{\pm0.5}$ |
| AI | (Wang et al., 2023a) 49.0 | $32.3_{\pm0.8}$ | $59.1_{\pm1.1}$ | $56.7_{\pm3.0}$ | $\mathbf{61.6}_{\pm1.9}$ |
| Literature | (Wang et al., 2023a) 47.2 | $39.4_{\pm0.7}$ | $\mathbf{62.7}_{\pm3.2}$ | $59.7_{\pm0.3}$ | $59.1_{\pm2.6}$ |
| Music | (Wang et al., 2023a) 53.2 | $56.2_{\pm1.3}$ | $67.8_{\pm0.2}$ | $65.5_{\pm3.6}$ | $\mathbf{68.4}_{\pm2.1}$ |
| Politics | (Wang et al., 2023a) 48.2 | $38.3_{\pm1.1}$ | $57.2_{\pm1.0}$ | $54.4_{\pm4.1}$ | $\mathbf{60.2}_{\pm3.0}$ |
| Science | (Wang et al., 2023a) 49.3 | $37.1_{\pm1.3}$ | $55.5_{\pm1.6}$ | $56.2_{\pm1.0}$ | $\mathbf{56.3}_{\pm0.4}$ |
| E3C | - | $59.8_{\pm0.3}$ | $59.0_{\pm0.7}$ | $59.0_{\pm0.8}$ | $\mathbf{60.0}_{\pm0.4}$ |
| FabNER | - | $06.1_{\pm0.4}$ | $24.8_{\pm0.6}$ | $25.4_{\pm0.5}$ | $\mathbf{26.3}_{\pm0.4}$ |
| HarveyNER | - | $23.2_{\pm0.4}$ | $37.3_{\pm1.8}$ | $\mathbf{41.3}_{\pm0.8}$ | $38.9_{\pm0.5}$ |
| Movie | (Wang et al., 2023a) 63.0 | $43.4_{\pm1.1}$ | $\mathbf{63.0}_{\pm0.6}$ | $62.5_{\pm1.0}$ | $62.4_{\pm1.4}$ |
| Restaurants | (Wang et al., 2023a) 21.0 | $31.3_{\pm2.2}$ | $43.4_{\pm0.8}$ | $49.8_{\pm1.4}$ | $\mathbf{52.7}_{\pm1.6}$ |
| MultiNERD | - | $55.0_{\pm1.1}$ | $76.0_{\pm0.7}$ | $\mathbf{77.5}_{\pm0.3}$ | $77.2_{\pm0.6}$ |
| $WikiEvents_{NER}$ | (Sainz et al., 2022b) *49.1 | $76.9_{\pm5.1}$ | $80.7_{\pm0.7}$ | $80.2_{\pm0.7}$ | $\mathbf{81.3}_{\pm0.5}$ |
| $WikiEvents_{EE}$ | (Sainz et al., 2022b) *10.4 | $47.5_{\pm0.4}$ | $43.0_{\pm0.6}$ | $45.7_{\pm0.8}$ | $\mathbf{47.0}_{\pm1.9}$ |
| $WikiEvents_{EAE}$ | Sainz et al. (2022a) 35.9 | $51.6_{\pm0.5}$ | $51.9_{\pm0.4}$ | $\mathbf{52.5}_{\pm1.2}$ | $50.7_{\pm0.4}$ |
| Average SoTA | 42.6 | $45.4_{\pm0.5}$ | $58.4_{\pm0.5}$ | $58.3_{\pm0.7}$ | $\mathbf{60.0}_{\pm1.0}$ |
| Average all | - | $42.3_{\pm0.2}$ | $55.3_{\pm0.2}$ | $56.0_{\pm0.2}$ | $\mathbf{57.2}_{\pm0.5}$ |

of 13 F1 points on average. Despite dividing the evaluation benchmarks based on the domain, there is always some overlap between labels of train and evaluation benchmarks. For instance, the datasets E3C and WikiEvents share a large part of their schema with datasets like BC5CDR, ACE05, and RAMS. This phenomenon is reflected in the results.

GoLLIE surpasses by a large margin the current zero-shot SoTA methods Instruct-UIE (Wang et al., 2023a) and Entailment based IE (Sainz et al., 2022b). Compared to Instruct-UIE, the main differences are the backbone model, the amount of training data, and, the use or not of the guidelines. Instruct-UIE leverages the 11B FlanT5 which is a T5 fine-tuned on 473 NLP datasets. With respect to the data, Instruct-UIE leverages a total of 34 IE datasets (counting different tasks as datasets) from diverse domains, we only leverage 12 datasets. Contrary to our method they do not use guideline information. Still, our method performs significantly better suggesting that the guidelines have an important effect on the results.

PromptNER (Zhang et al., 2023a) also adds some definition information into the prompt in order to perform zero-shot NER. We compare our approach with them (represented as GPT-3.5) in Figure 1. Although their approach leverages guidelines too, our approach performs significantly better on all datasets, showing that LLMs (even with 175B parameters) struggle to follow guidelines. They solve this by adding examples in the context but are still far behind on a comparable setting (T5-XXL).

**Seen vs unseen labels:** Not all labels in the zero-shot datasets are unseen; there is an overlap between the labels in the training and zero-shot datasets. Although these labels may have very different annotation guidelines, we also report results on the set of labels to which it has not been exposed during training, to better understand the generalization capabilities of GoLLIE. The list of seen and unseen labels, as well as an extended analysis is available in Appendix B. Figure 4 aggregates the F1 scores across datasets for seen and unseen labels in the zero-shot scenario. All models exhibit slightly lower performance on unseen labels. For the baseline model, the performance drop is more pronounced. In contrast, GoLLIE demonstrates better generalization ability, showing a smaller gap in F1 scores between the seen and unseen labels. Also, the gap is smaller as the parameter count of our model increases.

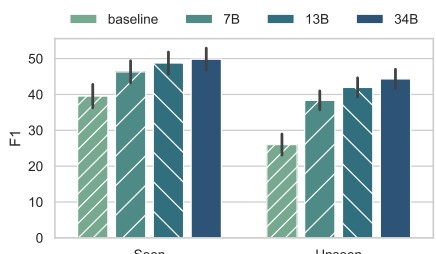

Figure 4: Seen vs unseen label zero-shot performance, results aggregated from all datasets.

**Model scaling:** Recent research has shown that increasing the parameter count of language models leads to improved generalization capabilities Brown et al. (2020). Higher parameter count yields superior average zero-shot performance. However, some datasets and tasks greatly benefit from a larger LLM, while others do not. We believe that some datasets do not see benefits from increasing the LLM size because their performance is hindered by the issues with the guidelines that we discuss in Section 5.3. While, in general, larger models achieve better results in both supervised and zero-shot settings, GoLLIE with a 7B parameter backbone already exhibits strong zero-shot capabilities.

## 5.3 ABLATION STUDY

We have performed an ablation to see the contribution of several components in the zero-shot evaluation. We analyzed the different regularization techniques proposed in Section 3.3. Additionally, we represent the baseline, i.e. when removing all components including guidelines, as "w/o *all*". Along with the mean zero-shot F1 we also provide the one-sided p-value with respect to 🐴 GoLLIE.

The class order shuffling, guideline paraphrasing, and class name masking seem to have no significant contribution to the final result, while class dropout although significant improvements are small. As further explained in Appendix D, the loss is only computed over the result tokens, inherently limiting the model's potential to overfit to the guidelines. In contrast, the representative annotation items give a stronger signal to the model. We see how definitions and representative candidates from the guidelines are complementary and help to improve each other.

Table 4: Ablation results.

| Model | F1 | p-value |
|---|---|---|
| 🐴 GoLLIE | $55.3_{\pm 0.2}$ | - |
| w/o Shuffling | $55.9_{\pm 0.2}$ | $7.2e^{-2}$ |
| w/o Paraphrases | $54.8_{\pm 0.2}$ | $1.1e^{-1}$ |
| w/o Masking | $54.6_{\pm 0.6}$ | $1.0e^{-1}$ |
| w/o Dropout | $54.0_{\pm 0.2}$ | $4.0e^{-3}$ |
| w/o Candidates | $49.9_{\pm 0.2}$ | $2.2e^{-10}$ |
| w/o *all* (baseline) | $42.3_{\pm 0.1}$ | $5.1e^{-13}$ |

## 6 ERROR ANALYSIS

In this section, we aim to better understand the effect of prompting LLMs with guidelines. We focus on specific labels across various datasets, with the results displayed in Table 5. Our analysis covers both successful and unsuccessful cases of entity labeling by GoLLIE. For the latter, we also aim to identify the reasons why the model fails to correctly label these entities. Further analyses on malformed outputs or hallucinations are discussed in Appendix C.

Table 5: This table shows the F1 scores for specific labels from different datasets. The guideline column is a small summary of the actual guideline used to prompt the model.

| Dataset | Label | Guideline | Baseline | 🐴 |
|---|---|---|---|---|
| MultiNERD | Media | Titles of films, books, songs, albums, fictional characters and languages. | 13.6 | 69.1 |
| CASIE | Vul. Patch | When a software company addresses a vulnerability by releasing an update. | 27.7 | 70.5 |
| Movie | Trailer | Refers to a short promotional video or preview of a movie. | 00.0 | 76.4 |
| AI | Task | Particular research task or problem within a specific AI research field. | 02.7 | 63.9 |
| MultiNERD | Time | Specific and well-defined time intervals, such as eras, historical periods, centuries, years and important days. | 01.4 | 03.5 |
| Movie | Plot | Recurring concept, event, or motif that plays a significant role in the development of a movie. | 00.4 | 05.1 |
| AI | Misc | Named entities that are not included in any other category. | 01.1 | 05.2 |
| Literature | Misc | Named entities that are not included in any other category. | 03.7 | 30.8 |
| Literature | Writer | Individual actively engaged in the creation of literary works. | 04.2 | 65.1 |
| Literature | Person | Person name that is not a writer. | 33.5 | 49.4 |
| Science | Scientist | A person who is studying or has expert knowledge of a natural science field. | 02.1 | 05.8 |
| Science | Person | Person name that is not a scientist. | 46.1 | 45.9 |
| Politics | Polit. Party | Organization that compete in a particular country's elections. | 11.2 | 34.9 |

**The details are in the guidelines:**   Labels such as MEDIA, VULNERABILITYPATCH, TRAILER, and TASK are inherently polysemous, making it challenging to determine the appropriate categorization based solely on the label name. As a result, the baseline struggles to effectively classify items under these labels due to having insufficient information. Conversely, GoLLIE successfully follows the guidelines, underscoring their utility.

**When the annotations do not comply with the guidelines:**   In the case of the TIME label of the MultiNERD dataset, we found that our model labels years as TIME entities. This is correct according to the annotation guidelines. Surprisingly, years are not labeled as entities in the dataset. In this case, GoLLIE successfully follows the guidelines; unfortunately, the dataset annotations do not.

**Ambiguous labels:**   The MISCELLANEOUS category, used by CoNLL03 and CrossNER datasets, refers to any named entity that is not included in the predefined categories set by the dataset. This definition is highly ambiguous and serves as a catch-all for various elements that do not fit into any of the predefined categories. Similarly, the PLOT category of the Movie dataset is used to label a wide range of elements. For example, events in a movie (e.g., murder, horse racing), characters (e.g., vampires, zombies), and the country of origin (e.g., British), among others. This lack of specificity hinders the development of consistent rules or guidelines for tagging such elements (Ratinov & Roth, 2009), which is a problem for humans and machines alike. As a consequence, GoLLIE also fails to label them accurately.

**Conflicts Between Fine-Grained and Coarse Entities:**   The CrossNER dataset introduces two labels for person names within each domain. For example, in the Science domain, the labels SCIENTIST and PERSON are used. The former is used to label any person that is not a Scientist. Similarly, the Literature domain includes the labels WRITER and PERSON. The guidelines assist GoLLIE in correctly labeling entities as WRITER. However, GoLLIE still categorizes individuals as *Person* even when they are *Scientist*, despite the guidelines. This is not technically incorrect, as every scientist is, by definition, also a person.

**Strong Label Preconceptions:**   In its Political domain set, CrossNER includes the label POLITICAL PARTY. GoLLIE outperforms the baseline, once again demonstrating the utility of providing the model with guidelines. However, we often find that the model categorizes political parties as organizations. As listed in Table 1, most of the pre-training datasets originate from the news domain, where political parties are a common entity. However, none of the fine-tuning datasets include the POLITICAL PARTY entity; they are instead categorized as ORGANIZATION. Consequently, during inference, the model consistently labels political parties as organizations. We believe this issue can be resolved by expanding the number and diversity of the fine-tuning datasets.

In summary, we anticipate that **GoLLIE will perform well on labels with well-defined and clearly bounded guidelines**. On the other hand, ambiguous labels or very coarse labels pose challenges. In this regard, we believe that GoLLIE would benefit from learning to follow instructions such as *"Label always the most specific class"* or *"Annotate this class in the absence of other specific class"*. We also expect that GoLLIE would benefit from expanding the number and diversity of the pre-training datasets.

## 7 CONCLUSIONS

In this paper, we introduce 🦙 GoLLIE, an LLM specifically fine-tuned to comply with annotation guidelines that were devised to help humans annotate the dataset. A comprehensive zero-shot evaluation empirically demonstrates that annotation guidelines are of great value for LLMs, as GoLLIE successfully leverages them. GoLLIE achieves better zero-shot results than previous attempts at zero-shot IE which do not leverage the guidelines, or use models not finetuned for following guidelines.

GoLLIE is a significant progress towards the development of models that can generalize to unseen IE tasks. In the future, we plan to enhance GoLLIE by using a larger and more diverse set of pre-training datasets. We will also improve the model's performance with ambiguous and coarse labels by expanding the set of instructions that the model can follow.

## ACKNOWLEDGMENTS

This work has been partially supported by the Basque Government (Research group funding IT-1805-22 and ICL4LANG project, grant no. KK-2023/00094). We are also thankful to several MCIN/AEI/10.13039/501100011033 projects: (i) DeepKnowledge (PID2021-127777OB-C21) and by FEDER, EU; (ii) Disargue (TED2021-130810B-C21) and European Union NextGenerationEU/PRTR; (iii) AWARE (TED2021-131617B-I00) and European Union NextGenerationEU/PRTR. This work has also been partially funded by the LUMINOUS project (HORIZON-CL4-2023-HUMAN-01-21-101135724). Oscar Sainz is supported by a doctoral grant from the Basque Government (PRE_2023_2_0137). Rodrigo Agerri currently holds the RYC-2017-23647 fellowship (MCIN/AEI/10.13039/501100011033 and by ESF Investing in your future).

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

```python
@dataclass
class Launcher(Template):
    """"Refers to a vehicle designed primarily to transport payloads from the Earth's
    surface to space. Launchers can carry various payloads, including satellites,
    crewed spacecraft, and cargo, into various orbits or even beyond Earth's orbit.
    They are usually multi-stage vehicles that use rocket engines for propulsion."""

    mention: str
    """
    The name of the launcher vehicle.
    Such as: "Sturn V", "Atlas V", "Soyuz", "Ariane 5"
    """
    space_company: str # The company that operates the launcher. Such as: "Blue origin", "ESA", "Boeing"
    crew: List[str]
    """"Names of the crew members boarding the Launcher.
    Such as: "Neil Armstrong", "Michael Collins", "Buzz Aldrin"
    """

@dataclass
class Mission(Template):
    """"Any planned or accomplished journey beyond Earth's atmosphere with specific objectives,
    either crewed or uncrewed. It includes missions to satellites, the International
    Space Station (ISS), other celestial bodies, and deep space."""

    mention: str
    """
    The name of the mission.
    Such as: "Apollo 11", "Artemis", "Mercury"
    """
    date: str # The start date of the mission
    departure: str # The place from which the vehicle will be launched. Such as: "Florida", "Houston"
    destination: str # The place or planet to which the launcher will be sent. Such as "Moon", "low-orbit"

# This is the text to analyze
text = (
    "The Ares 3 mission to Mars is scheduled for 2032. The Starship rocket build by SpaceX will take off"
    "from Boca Chica, carrying the astronauts Max Rutherford, Elena Soto, and Jake Martinez."
)

# The annotation instances that take place in the text above are listed here
result = [
    Mission(mention='Ares 3', date='2032', departure='Boca Chica', destination='Mars'),
    Launcher(mention='Starship', space_company='SpaceX', crew=['Max Rutherford', 'Elena Soto', 'Jake Martinez'])
]
```

Figure 5: Example of generalization to custom tasks defined by the user.

## A    EXAMPLES

In addition to NER, EE and EAE for which examples are shown in Figures 2 and 3 respectively, we also feed the model with data from RE and SF. The formulation of RE is similar to the NER but with two argument attributes. However, the SF task is more complex as shown in Figure 6. With this task, we added several layers of complexity to the input: extended definitions for each possible attribute (slot), optional arguments, and, fine-grained definitions of types such as Names, Values, or Strings. We also added constraints into the prompt to condition the model to just output the information of our desired query instead of every single template on the text. In the future, we would like to add more complex tasks to the training and evaluation to improve the capabilities and flexibility of the model. For more examples refer to the GitHub repository.

### A.1    EXAMPLE OF GENERALIZATION TO NEW CUSTOM TASKS

Our model allows the user to define custom annotation schemas using Python code. We provide an example where we define two new types of entities: Launcher and Mission. As shown in Figure 5, Launcher and Mission are not simple entities, they correspond to what we call *Template*, a class similar to *Entity* but with additional arguments, like the SF task. For example, the space_company or the crew of the launcher are some of the additional arguments we added to the schema. As shown in the example, the model's output (everything after `result = []`) satisfies the type constraints defined in the guidelines, attributes defined as strings are filled with strings and, the arguments defined as lists (like *crew*) are filled with lists. The model can correctly analyze the given sentence with our newly created annotation schema.

```
# The following lines describe the task definition        # This is the text to analyze
@dataclass                                                text = "Mongolian Prime Minister M. Enkhbold met
class PersonTemplate(Template):                            with Liu Hongcai , vice minister of the
    """Person templates encodes the information about the given query    International Department of the Chinese Communist
    Person entity."""                                     Party Central Committee on Monday here ."

    query: str  # The Person entity query                 # The list called result contains the templates
    alternate_names: Optional[List[Name]] = None          # instances for the following entity queries:
    """Names used to refer to the query person that are distinct from the   #    - M. Enkhbold: PersonTemplate
    'official' name. Including: aliases, stage names, abbreviations ..."""   #
    date_of_birth: Optional[Value] = None                 result = [
    """The date on which the query person was born."""        PersonTemplate(
    age: Optional[Value] = None                                   query="M. Enkhbold",
    """A reported age of the query person."""                     countries_of_residence=[Name("Mongolian")],
    city_of_birth: Optional[Name] = None                          title=[String("Prime Minister")],
    """The geopolitical entity at the municipality level (city, town, or      ),
    village) in which the query person was born"""            ]
    date_of_death: Optional[Value] = None
    """The date of the query person's death."""

                    (Collapsed 36 more slots)
```

Figure 6: Example of the TACRED dataset converted to Slot Filling task represented as code.

## B    PERFORMANCE IN SEEN VS UNSEEN LABELS: FURTHER ANALYSIS

Table 6: List of labels in the zero-shot datasets that overlap with the ones in the training datasets (seen) and the labels that do not overlap with the ones in the training datasets (unseen)

| Dataset | Seen Labels | Unseen Labels |
|---|---|---|
| BroadTwitter | Location, Organization, Person | - |
| CASIE$_{EE}$ | - | DatabreachAttack, PhisingAttack, RansomAttack, VulnerabilityDiscover, VulnerabilityPatch |
| AI | Product, Country, Person, Organization, Location, Miscellaneous | Field, Task, Algorithm, Researcher, Metric, University, ProgrammingLanguage, Conference |
| Literature | Event, Person, Location, Organization, Country, Miscellaneous | Book, Writer, Award, Poem, Magazine, LiteraryGenre |
| Music | Event, Country, Location, Organization, Person, Miscellaneous | MusicGenre, Song, Band, Album, MusicalArtist, MusicalInstrument, Award |
| Politics | Person, Organization, Location, Election, Event, Country, Miscellaneous | Politician, PoliticalParty |
| Science | Person, Organization, Country, Location, ChemicalElement, ChemicalCompound, Event, Miscellaneous | Scientist, University, Discipline, Enzyme, Protein, AstronomicalObject, AcademicJournal, Theory, Award |
| E3C | ClinicalEntity | - |
| FabNER | Biomedical | Material, ManufacturingProcess, MachineEquipment, Application, EngineeringFeatures, MechanicalProperties, ProcessCharacterization, ProcessParameters, EnablingTechnology, ConceptPrinciples, ManufacturingStandards |
| HarveyNER | - | Point, Area, Road, River |
| Movie | Year | Actor, Character, Director, Genre, Plot, Rating, RatingsAverage, Review, Song, Tittle, Trailer |
| Restaurants | Location, Price, Hours | Rating, Amenity, RestaurantName, Dish, Cuisine |
| MultiNERD | Person, Location, Organization, Biological, Disease, Event, Time, Vehicle | Animal, Celestial, Food, Instrument, Media, Plant, Mythological |
| WikiEvents$_{NER}$ | CommercialProduct, Facility, GPE, Location, MedicalHealthIssue, Money, Organization, Person, JobTitle, Numeric, Vehicle, Weapon | Abstract, BodyPart, Information, SideOfConflict |
| WikiEvents$_{EE}$ | ConflictEvent, ContactEvent, GenericCrimeEvent, JusticeEvent, MedicalEvent, MovementTransportEvent, PersonnelEvent, TransactionEvent | ArtifactExistanceEvent, CognitiveEvent, ControlEvent, DisasterEvent, LifeEvent |

Table 6 categorizes the labels for each zero-shot dataset into those that overlap with the training dataset and those that are completely unseen. We adhere to a strict approach in this classification. For instance, although the label COUNTRY does not appear in the training datasets, similar labels such as GEOPOLITICAL entity do. Therefore, we consider that the model has been exposed to this label during training.

While some labels in the zero-shot datasets overlap with those in the training dataset, the annotation guidelines for each label may vary significantly between datasets. Table 7 presents the micro-F1

Table 7: Micro F1 score for the seen and unseen labels in the zero-shot datasets.

| Dataset | Baseline Seen | Baseline Unseen | 🐻 Seen | 🐻 Unseen | 🐻 13B Seen | 🐻 13B Unseen | 🐻 34B Seen | 🐻 34B Unseen |
|---|---|---|---|---|---|---|---|---|
| BroadTwitter | $39.0_{\pm0.6}$ | - | $49.5_{\pm0.8}$ | - | $51.4_{\pm1.8}$ | - | $50.3_{\pm2.1}$ | - |
| CASIE$_{EE}$ | - | $33.9_{\pm6.5}$ | - | $59.3_{\pm2.3}$ | - | $62.2_{\pm0.9}$ | - | $65.5_{\pm1.8}$ |
| AI | $43.5_{\pm1.4}$ | $21.1_{\pm0.3}$ | $57.8_{\pm1.2}$ | $60.0_{\pm1.2}$ | $57.8_{\pm0.8}$ | $55.8_{\pm4.7}$ | $57.7_{\pm2.8}$ | $64.2_{\pm1.3}$ |
| Literature | $34.6_{\pm0.2}$ | $43.6_{\pm1.5}$ | $54.6_{\pm3.6}$ | $67.4_{\pm3.0}$ | $52.4_{\pm0.2}$ | $64.6_{\pm0.5}$ | $52.7_{\pm2.1}$ | $63.7_{\pm2.8}$ |
| Music | $46.8_{\pm1.0}$ | $62.2_{\pm1.6}$ | $53.7_{\pm0.2}$ | $74.9_{\pm0.3}$ | $52.8_{\pm3.9}$ | $72.7_{\pm3.5}$ | $54.0_{\pm3.8}$ | $76.3_{\pm1.2}$ |
| Politics | $45.9_{\pm1.1}$ | $4.6_{\pm2.6}$ | $64.0_{\pm0.2}$ | $31.9_{\pm4.7}$ | $62.0_{\pm2.2}$ | $22.4_{\pm14.6}$ | $64.4_{\pm1.5}$ | $45.8_{\pm9.3}$ |
| Science | $38.7_{\pm0.8}$ | $34.7_{\pm3.0}$ | $52.7_{\pm1.7}$ | $58.8_{\pm1.5}$ | $52.7_{\pm1.0}$ | $60.4_{\pm0.9}$ | $52.5_{\pm0.4}$ | $60.5_{\pm0.7}$ |
| E3C | $59.8_{\pm0.3}$ | - | $59.0_{\pm0.7}$ | - | $59.0_{\pm0.9}$ | - | $60.0_{\pm0.4}$ | - |
| FabNER | $0.0_{\pm0.0}$ | $6.2_{\pm0.4}$ | $22.6_{\pm2.3}$ | $24.9_{\pm0.6}$ | $23.9_{\pm4.4}$ | $25.5_{\pm0.6}$ | $20.7_{\pm2.9}$ | $26.5_{\pm0.6}$ |
| HarveyNER | - | $23.2_{\pm0.4}$ | - | $37.3_{\pm1.8}$ | - | $41.3_{\pm0.9}$ | - | $38.9_{\pm0.5}$ |
| Movie | $31.5_{\pm0.7}$ | $46.1_{\pm1.5}$ | $58.7_{\pm2.3}$ | $63.8_{\pm0.5}$ | $47.3_{\pm3.1}$ | $65.3_{\pm1.0}$ | $42.7_{\pm2.3}$ | $66.1_{\pm1.4}$ |
| Restaurants | $18.0_{\pm1.1}$ | $38.7_{\pm2.8}$ | $33.2_{\pm2.7}$ | $49.9_{\pm1.5}$ | $38.0_{\pm3.6}$ | $57.1_{\pm0.2}$ | $46.0_{\pm4.2}$ | $57.2_{\pm0.9}$ |
| MultiNERD | $58.0_{\pm1.1}$ | $39.5_{\pm1.4}$ | $81.2_{\pm0.5}$ | $44.6_{\pm0.9}$ | $82.4_{\pm0.4}$ | $47.7_{\pm0.7}$ | $82.3_{\pm0.5}$ | $49.1_{\pm0.5}$ |
| WikiEvents$_{NER}$ | $77.2_{\pm5.1}$ | $0.0_{\pm0.0}$ | $81.5_{\pm0.7}$ | $0.0_{\pm0.0}$ | $80.9_{\pm0.8}$ | $0.0_{\pm0.0}$ | $82.1_{\pm0.5}$ | $3.5_{\pm2.6}$ |
| WikiEvents$_{EE}$ | $43.3_{\pm0.3}$ | $57.2_{\pm1.5}$ | $41.7_{\pm0.1}$ | $45.0_{\pm1.5}$ | $43.9_{\pm0.8}$ | $48.8_{\pm1.7}$ | $45.0_{\pm1.1}$ | $50.4_{\pm0.9}$ |
| Average | $41.2_{\pm0.4}$ | $31.6_{\pm0.6}$ | $54.6_{\pm0.3}$ | $47.5_{\pm0.6}$ | $54.2_{\pm0.4}$ | $48.0_{\pm1.3}$ | $54.7_{\pm0.9}$ | $51.4_{\pm1.0}$ |

scores for both seen and unseen labels across each zero-shot dataset. Generally, the models perform better on seen labels compared to unseen ones. However, there are instances where the reverse is true. This occurs when a dataset contains labels that, although overlapping with those in the training dataset, have vastly different annotation guidelines. As discussed in Section 5.3, the model has strong preconceptions for some labels, adversely affecting zero-shot performance. GoLLIE, trained to adhere to specific annotation guidelines, demonstrates greater robustness against these label preconceptions than the baseline model. Consequently, it achieves better results for both seen and unseen labels. GoLLIE can successfully handle both, seen and unseen labels from datasets that were not used during training. This ability underscores GoLLIE's superior generalization capabilities, largely attributable to its capability of leveraging annotation guidelines.

## C  MODEL HALLUCINATIONS

Table 8: Number impossible to parse outputs and number predicted labels that are hallucinations. F1 scores on the dataset are shown for reference.

| Dataset | 🐻 Impossible to Parse | Hallucinations | F1 Score |
|---|---|---|---|
| BroadTwitter | $0_{\pm0}$ / 2002 | $0_{\pm0}$ / 1664 | $49.5_{\pm0.8}$ |
| CASIE$_{EE}$ | $1_{\pm0}$ / 199 | $6_{\pm1}$ / 1548 | $59.3_{\pm2.3}$ |
| CASIE$_{EAE}$ | $1_{\pm1}$ / 199 | $3_{\pm2}$ / 2804 | $50.0_{\pm1.1}$ |
| AI | $0_{\pm0}$ / 431 | $1_{\pm1}$ / 1292 | $59.1_{\pm1.1}$ |
| Literature | $0_{\pm0}$ / 416 | $0_{\pm0}$ / 2059 | $62.7_{\pm3.2}$ |
| Music | $0_{\pm0}$ / 465 | $6_{\pm2}$ / 3080 | $67.8_{\pm0.2}$ |
| Politics | $0_{\pm0}$ / 651 | $3_{\pm2}$ / 4142 | $57.2_{\pm1.0}$ |
| Science | $0_{\pm0}$ / 543 | $7_{\pm1}$ / 2700 | $55.5_{\pm1.6}$ |
| E3C | $0_{\pm0}$ / 851 | $1_{\pm0}$ / 688 | $59.0_{\pm0.7}$ |
| FabNER | $1_{\pm0}$ / 2064 | $13_{\pm3}$ / 4474 | $24.8_{\pm0.6}$ |
| HarveyNER | $0_{\pm0}$ / 1303 | $1_{\pm1}$ / 708 | $37.3_{\pm1.8}$ |
| Movie | $0_{\pm0}$ / 2443 | $1_{\pm0}$ / 3919 | $63.0_{\pm0.6}$ |
| Restaurants | $0_{\pm0}$ / 1521 | $3_{\pm0}$ / 1451 | $43.4_{\pm0.8}$ |
| MultiNERD | $49_{\pm11}$ / 32908 | $51_{\pm8}$ / 67142 | $76.0_{\pm0.7}$ |
| WikiEvents$_{NER}$ | $0_{\pm0}$ / 573 | $1_{\pm1}$ / 2666 | $80.7_{\pm0.7}$ |
| WikiEvents$_{EE}$ | $0_{\pm0}$ / 573 | $3_{\pm1}$ / 630 | $43.0_{\pm0.6}$ |
| WikiEvents$_{EAE}$ | $2_{\pm1}$ / 321 | $0_{\pm0}$ / 363 | $51.9_{\pm0.4}$ |

In this section, we evaluate the hallucinations generated by the model. We examine two different phenomena. First, we consider instances where the output is so corrupted that it is *impossible to parse*. In such cases, we treat the output as an empty list. Second, we look at instances where the model outputs a label *Hallucination*, that is, a label not defined among the input classes. In these instances, we remove the label from the output. As demonstrated in Table 8, for all the zero-shot datasets, both phenomena occur in less than 1% of the predictions. This demonstrates that GoLLIE is highly resistant to hallucinations and closely adheres to the classes defined in the input.

# D EXTENDED TRAINING DETAILS

## D.1 LOSS CALCULATION

We have used the standard Next Token Prediction (NTP) loss to train our models. However, several regularizations that we applied to the models made the loss computed over the guideline tokens much higher than the actual output tokens. This is because we randomly shuffle the guidelines order, mask names, or, drop classes, which makes it impossible to predict what goes next. To avoid the loss of the guideline tokens overshadowing the actual output tokens loss, we decided to only compute the loss over the output tokens. This way, we can also avoid some overfitting of the guidelines. This resulted in faster training and better results overall.

## D.2 DATASET DETAILTS

Table 9 shows the number of examples for each training and zero-shot dataset. OntoNotes is generated semi-automatically and is orders of magnitude larger than the other ones. Therefore, for each training epoch, we sample 30.000 random examples from the training set. The models were trained for 3 epochs with an effective batch size of 32 and a learning rate of 3e-4 with a cosine scheduler. Therefore, we perform 15,485 training steps.

Regarding the splits, we use the standard train, dev test splits for every dataset. In the case of ACE, we follow the split provided by Lin et al. (2020). In the case of CASIE, we took the first 200 instances as validation and the last 2000 as test.

Table 9: Number of examples for each training and zero-shot dataset

| Dataset | Train | Dev | Test |
|---|---|---|---|
| ACE05$_{NER}$ | 19217 | 676 | 901 |
| ACE05$_{RE}$ | 19217 | 901 | 676 |
| ACE05$_{EE}$ | 19217 | 676 | 901 |
| ACE05$_{EAE}$ | 3843 | 397 | 368 |
| ACE05$_{RC}$ | 5691 | - | - |
| ACE05$_{VER}$ | 19217 | - | - |
| BC5CDR | 4561 | 4582 | 4798 |
| CoNLL 2003 | 14041 | 3250 | 3453 |
| DIANN | 3976 | 793 | 1309 |
| NCBIDisease | 5433 | 924 | 941 |
| Ontonotes 5 | *30000* | 15680 | 12217 |
| RAMS | 7329 | 924 | 871 |
| TACRED | 10027 | 3896 | 2311 |
| WNUT 2017 | 3394 | 1009 | 1287 |
| Total | 165163 | 33708 | 30033 |
| BroadTwitter | - | - | 2002 |
| CASIE$_{EE}$ | - | - | 199 |
| CASIE$_{EAE}$ | - | - | 199 |
| AI | - | - | 431 |
| Literature | - | - | 416 |
| Music | - | - | 465 |
| Politics | - | - | 651 |
| Science | - | - | 543 |
| E3C | - | - | 851 |
| FabNER | - | - | 2064 |
| HarveyNER | - | - | 1303 |
| Movie | - | - | 2443 |
| Restaurants | - | - | 1521 |
| MultiNERD | - | - | 32908 |
| WikiEvents$_{NER}$ | - | - | 573 |
| WikiEvents$_{EE}$ | - | - | 573 |
| WikiEvents$_{EAE}$ | - | - | 321 |
| Total | - | - | 47463 |

Table 10: Details about the training resources required for each model.

| Model | Hardware | FLOPs | Time (h) | $CO_2$eq (kg) |
|---|---|---|---|---|
| Baseline | 1xA100 | $4.5e^{18}$ | 17.3 | 0.61 |
| 🐻 GoLLIE | 1xA100 | $11.9e^{18}$ | 44.5 | 1.57 |
| 🐻 13B | 1xA100 | $22.7e^{18}$ | 79.5 | 2.80 |
| 🐻 34B | 2xA100 | $55.8e^{18}$ | 94.6 | 6.67 |

## D.3 CARBON FOOTPRINT

Fine-tuning LLMs is not as expensive as pre-training these models. Still, we believe that it is important to measure and document the costs that our experiments have on our planet. We provide the resources required for a single run of our experiments in Table 10. All the experiments were done on our private infrastructure. For the carbon footprint estimation, we estimated the values considering a 400W consumption per GPU with a 0.141 kg/kWh carbon intensity[†].

## D.4 LORA VS FULL MODEL FINE-TUNING

We conducted preliminary experiments to compare the performance of QLoRA (Hu et al., 2022; Dettmers et al., 2023) with that of training all the parameters in the model. These preliminary experiments were conducted using the LLaMA2 7B model Touvron et al. (2023b) and an early version of the code. However, the experimental setup for both models was identical. Both approaches were prompted with guidelines. First, we compared the training loss of both approaches. Figure 7 shows that when fine-tuning all the parameters, the loss decreases much more rapidly than when training only the LoRA layers. It also achieves a lower loss at the end of training. However, when evaluating the model at the end of the first and third epochs, we observed that training the full model performs very poorly, as shown in Table 11. We hypothesize that when training all the parameters, the model overfits quickly (indicated by the lower training loss) and memorizes the training data. On the other hand, training only the LoRA layers, which represent around 0.5% of the total model weights, introduces a bottleneck that prevents the model from memorizing the training dataset. It is also noteworthy that the QLoRA approach was trained using just one Nvidia A100 80GB GPU thanks to the 4-bit quantization of the frozen model Dettmers et al. (2023). Training the full model required a minimum of four Nvidia A100 80GB GPUs to fit the model into memory. We used DeepSpeed [‡] to distribute the model across the four GPUs for training. Due to the high cost of training, we did not perform an extensive hyperparameter search for the full model.

Figure 7: Training loss of fine-tuning the full model vs training LoRA layers only.

---
[†]Statistic taken from `https://app.electricitymaps.com/map`
[‡]`github.com/microsoft/DeepSpeed`

Table 11: F1 scores achieved when training the full model vs only training the LoRA Layers at the end of the first and third epoch.

| Training | Epoch | Precision | LR | HarveyNer | FabNER | Restaurant | Movie | CASIE$_{EE}$ | CoNLL03 |
|---|---|---|---|---|---|---|---|---|---|
| Full | 1 | BF16 | 1e−4 | 0.00 | 0.00 | 0.25 | 4.74 | 0.00 | 85.57 |
| Full | 3 | BF16 | 1e−4 | 3.45 | 0.21 | **46.7** | 16.72 | 0.42 | 84.83 |
| QLoRA | 1 | 4Bit + BF16 | 2e−3 | 34.98 | **20.78** | 45.01 | **51.14** | 55.83 | 91.41 |
| QLoRA | 3 | 4Bit + BF16 | 2e−3 | **35.34** | 16.21 | 39.07 | 44.18 | **57.93** | **93.14** |

## E  HANDLING DATASETS WITH HUNDREDS OF LABELS AND CODE-STYLE PROMPT OVERHEAD

In our research, we focus on datasets with fewer than 20 labels. However, some datasets, such as FIGER Ling & Weld (2012), include hundreds of fine-grained labels. Including guidelines in datasets with hundreds of labels can make inputs excessively long, exceeding the context size of current Large Language Models (LLMs). This is a known constraint in LLMs, and recently, significant research effort has been directed towards algorithms that efficiently increase the context window size Press et al. (2022). We anticipate that future LLMs will have a context window large enough to accommodate not only more labels but also more detailed guidelines. For the time being, this problem can be mitigated by batching the labels into multiple input examples. Instead of prompting the model with, for example, 100 labels in a single input, it is possible to prompt the model with 10 inputs, each incorporating 10 labels, and then combine all the outputs into a single response. In any case, handling datasets with a large number of labels remains a limitation of GoLLIE.

Figure 8: Percentage of characters from the input required to represent the code-style prompt for different labels. For detailed guidelines, the code is a small fraction of the input.

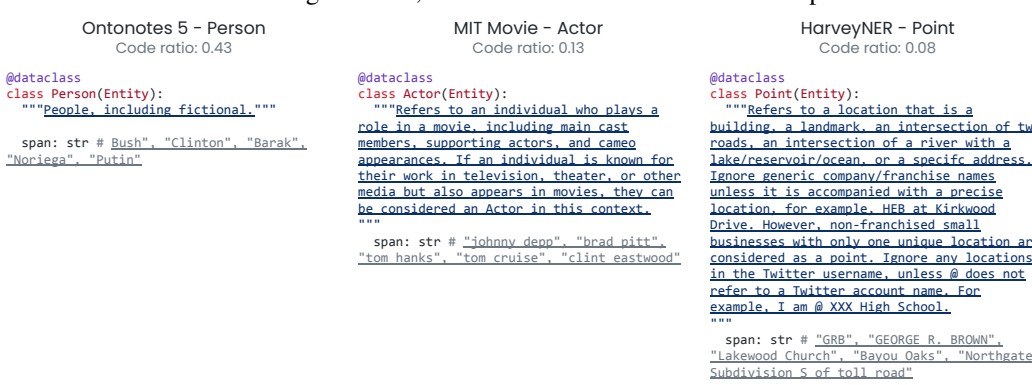

Our approach uses Python-based code-style prompts, this requires including tokens in the input to represent the code structures. Figure 8 illustrates various labels formatted in our code-style input alongside their respective guidelines. For very generic guidelines, such as those for the PERSON entity in OntoNotes 5, the code structure accounts for almost half of the input's characters. However, for detailed guidelines, like those for the POINT entity in HarveyNER, the code structure constitutes only a small portion of the input. While there is an overhead of tokens to represent the code structure, when dealing with datasets with a very large number of labels, the primary limitation is fitting the guideline definitions into the model's input, rather than accommodating the Python code structure.

## F    HUMAN EFFORT TO BUILD THE PROMPTS

GoLLIE requires formatting the input in a Python-based code representation. We achieve this by filling pre-defined templates for each task (NER, EE, EAE, RE, SF). We will make this predefined set of templates publicly available along with our code. Implementing new datasets only requires defining a list of labels and the guidelines for each label. We reuse the annotation guidelines provided by the dataset authors. Therefore, for most datasets, this process is straightforward and requires very little human effort. For datasets with very large and complex guidelines, such as TACRED, manual summarization of the guidelines was necessary. In any case, the involvement of a domain expert is not required. Additionally, since the inputs are automatically generated using templates, implementing new datasets does not require knowledge of Python coding.

Some datasets did not have publicly available guidelines, either due to semi-automatic generation or because the authors chose not to disclose them. For these specific datasets, human experts were needed to construct the guidelines from examples in the development split. We plan to release our generated guidelines to support future research. Human domain experts are necessary to adapt GoLLIE to new tasks where guidelines or annotations are unavailable. However, this requirement is common to any other Information Extraction (IE) model.

## G    DATA-CONTAMINATION STATEMENT

We believe data contamination is a relevant problem that affects the NLP evaluations nowadays, becoming more prevalent with LLMs (Dodge et al., 2021; Magar & Schwartz, 2022; Sainz et al., 2023b;a). Detecting whether a dataset was inside an LLM pre-trained corpora is challenging even with the pre-training data itself. In this paper, unfortunately, we do not have access to the pre-training data used to train Code-LLaMA the backbone LLM of our model. This issue is particularly worrying for us because one big source of contamination is probably GitHub and other code repositories which are also used to upload evaluation benchmarks Dodge et al. (2021). As Code-LLaMA is trained on code, there is a chance for this particular data leakage. However, all of our comparisons were made against our baseline, which has the same backbone LLM as GoLLIE. Even if the results were impacted, **the improvements of our model over the baseline would not be affected by data contamination as both share the same pre-training**. We hope that in the future more transparency will allow to perform safer evaluations.

