# OpenReview forum: "GoLLIE: Annotation Guidelines improve Zero-Shot Information-Extraction"
_ICLR.cc/2024/Conference — ICLR 2024 poster_

### Official Review · Reviewer_zWZU · 2023-10-29

**Soundness:** 3 good
**Presentation:** 3 good
**Contribution:** 3 good
**Rating:** 8
**Confidence:** 5

**Summary:**

This paper proposes GoLLIE, a large language model fine-tuned to follow label descriptions and few-shot examples to improve zero-shot information extraction. Key ideas include formulating tasks as Python code, regularizing training, and comprehensive evaluation on diverse IE tasks.

**Strengths:**

* LLMs have excelled at numerous complex tasks, but underperformed in IE tasks. This paper addresses this gap, diving deep into the annotation guidelines and error analysis using an LLM approach: zero-shot learning, natural language instructions, programming language output format, and leveraging one model to tackle multiple tasks.
* The study's thorough experiments span various IE tasks, coupled with an extensive error analysis when applying the unified model across these diverse tasks. Such detailed error analysis offers valuable insights into the model's strengths and limitations.
* Another significant contribution is crafting prompts for intricate IE tasks. Demonstrating the model's ability to adhere to these instructions, especially given the recognized challenges of current LLMs in following complex instructions, signifies substantial effort and expertise from the IE domain.

**Weaknesses:**

* The claim of making use of “annotation guideline” may be an overstatement - this paper only considered label name, label description and few-shot examples, however, annotation guideline in IE domain are very complicated and was curated by linguists. E.g., For TACRED slot filling (https://tac.nist.gov/2015/KBP/ColdStart/guidelines/TAC_KBP_2015_Slot_Descriptions_V1.0.pdf), section 3.6 per:city_of_birth, they use “GPEs below the city level (e.g. 5 boroughs of New York City) are not valid fillers.“ as an example rule to guide annotators. The prompts proposed by this paper might not fully capture the depth of true guideline understanding.
* The paper omits key references from the era before LLMs that discuss label descriptions and verbalization. Examples include:
    * Zero-Shot Relation Extraction via Reading Comprehension
    * An Empirical Study on Multiple Information Sources for Zero-Shot Fine-Grained Entity Typing
    * Label Verbalization and Entailment for Effective Zero- and Few-Shot Relation Extraction

Small formatting issues:
* Use ``’’ instead of ‘’’’.
* Table 6 could be more user-friendly. Presenting exact numbers (like 10/10,000) would be clearer than percentages.

Please see more in the question section.

**Questions:**

* How does the prompt length vary in the proposed method? Translating IE tasks into Python classes might make them more accessible for LLMs, but could also significantly inflate the token count. For instance, a fine-grained entity type classification task with over 1000 entity types would result in a considerable token overhead.
* The paper doesn't detail the human (or domain expert) effort required to craft the prompts. Given the variety of labels and tasks covered in the study, it would be beneficial to track this for subsequent research.

These are important questions that need to be answered in the updated paper.

---

> ### Author Response · Authors · 2023-11-17
>
> Regarding the first weakness, yes we completely agree on that. It is something that we definitely want to try. However those guidelines are very long, and considering the amount of labels we have per example, it is intractable with our current models/infrastructure setup. With recent efficient methods and models for larger sequences we hope we could soon explore the use of more detailed and longer guidelines in our experiments. We believe that the insights from this paper will motivate further research in this direction.
>
> Regarding the second weakness, we will happily add the missing references (see paper changes). In fact, one of them was already in the paper. Thank you for your suggestions.
>
> Thank you for your recommendations, we will update the paper accordingly.
>
> For the first question, we have added to the updated paper a new discussion section (Appendix E) related to the prompt length and scalability. However, answering your question, the amount of tokens required to represent the code structures is very small in contrast with the tokens used for the guidelines. This depends on how detailed the guidelines are and on the specific tokenizer. But as a reference, for representing the class “Actor” of the MitMovie named entity dataset, the code structure accounts for 52 characters, while the actual guideline requires 350 characters. In the HarverNER Point entity, which has a more extensive and detailed guideline, the code structure accounts for only 8% of the total characters of the guideline. While there is a small overhead of tokens, when dealing with a dataset with 1000 entity types, the main limitation would be fitting into the model input 1000 guideline definitions, not the Python code structure.
>
> Regarding your second question, the human effort involved was very little in the case of freely available guidelines. We just needed to copy and paste, and if they were very long and detailed (the case of ACE and TACRED for example), simplify. When no guidelines are provided for a given dataset, we either reuse some templates from previous works (for WikiEvents for example) or follow other dataset styles to generate very simple guidelines. The human effort here was also very little. The more important point is that no domain expert was required. We have added a new section in the updated paper (Appendix F) with a more detailed disclosure of the human effort involved to generate the inputs.

---

> > ### Comment · Reviewer_zWZU · 2023-11-20
> >
> > Thank you for addressing my concerns in the revised version of your paper.
> >
> > I would like to emphasize a particularly unique and impactful aspect of your work: the error analysis concerning label definition conflicts. This analysis is especially enlightening and holds significant implications for future research. In the current era of LLMs, resolving conflicts that arise from training a single model on multiple existing datasets represents a critical challenge. Your paper makes a substantial contribution in this area.
> >
> > I will maintain my original score for the paper.

---

### Official Review · Reviewer_xrh5 · 2023-10-29

**Soundness:** 2 fair
**Presentation:** 2 fair
**Contribution:** 3 good
**Rating:** 5
**Confidence:** 4

**Summary:**

The paper introduces GoLLIE (Guideline-following Large Language Model for IE), a model fine-tuned to effectively utilize annotation guidelines for enhanced zero-shot results in Information Extraction tasks. A primary distinction of GoLLIE is its use of a Python code-based representation for both input and output. This method provides a clear, human-readable format, unifying the representation of various IE tasks and addressing challenges associated with traditional natural language instructions. Empirical evaluations validate GoLLIE's superior capabilities in leveraging guidelines, indicating promising directions for future research and model enhancements, including the exploration of more extensive pre-training datasets and refining its response to ambiguous labels.

**Strengths:**

1. The paper presents a novel solution to the long-standing challenge where large models struggled to leverage intricate annotation guidelines inherently. Through targeted fine-tuning and a unified Python code-based representation for both input and output, it significantly improves zero-shot outcomes in IE tasks.

2. The detailed error analysis provides invaluable insights into the challenges of zero-shot IE, especially when leveraging guidelines. The study methodically identifies specific issues like the ambiguity of labels, conflicts between fine-grained and coarse entities, and the repercussions of strong label preconceptions. These findings offer a valuable roadmap for subsequent research in the domain, emphasizing the need for clearer guidelines, broader fine-tuning datasets, and more specific instruction-following capabilities.

3. The paper is well-written and easy to follow.

**Weaknesses:**

1. The paper's novelty appears somewhat limited as similar approaches have been explored in past research. [1] proposes a zero-shot entity typing approach that utilizes the type description available from Wikipedia to build a distributed semantic representation of the types.

2. The choice of code-based prompting can potentially hinder usability for general users. Those unfamiliar with coding might find it challenging to interact with or fully leverage the model, thereby narrowing its applicability.

3. The method's efficacy remains unclear due to incomplete experimental validation. Specific concerns include:

   a. **Prompt Style Impact**: The rationale for using a Python code style prompt is not experimentally validated. It's untested whether natural language prompts with guidelines might yield similar outcomes. Although CodeIE validated code style prompts for OpenAI models, the necessity of this design for open-source LLMs during instruction tuning remains unproven.

   b.**Prompt Sensitivity**: The paper omits experiments examining the sensitivity of prompts, leading to concerns about the method's stability. The model's performance under varying definitions, code structures, and code comment styles remains unexplored, making its robustness questionable.

4. The paper lacks experiments analyzing the impact of training dataset diversity on the LLM's ability to follow unseen guidelines. It remains unclear how much data is required to achieve satisfactory performance, leaving questions about the scalability and efficiency of the approach.



[1] Description-Based Zero-shot Fine-Grained Entity Typing

**Questions:**

1. **Sampling Strategy**: Datasets differ in scale, and a naive mixture could introduce imbalance. How did the authors handle the sampling process for the datasets used in this study? Were experiments conducted to assess the effects of different sampling techniques on the results? This aspect wasn't addressed in the paper and could provide clarity on the method's robustness across varied data distributions.
2. **Scalability with Numerous Labels**: For datasets with a large number of labels, say in the order of hundreds, the input length for your method could become substantially lengthy, leading to efficiency concerns. In such complex scenarios with multiple labels, how does the inclusion of guidelines impact the performance? This wasn't touched upon in the paper but would provide deeper insights into the method's scalability and effectiveness in real-world applications.
3. **QLoRA vs. Full Model Fine-tuning**: The authors mentioned employing QLoRA for training, citing its superior performance over fine-tuning the entire model on zero-shot tasks and faster training speed. However, the paper lacks empirical evidence to support this. What are the relative impacts of full-parameter SFT and techniques like LoRA and QLoRA on zero-shot IE performance?
4. **Benchmark Selection**: Many of the state-of-the-art methods used for comparison in the paper, such as UIE, appear outdated. Why weren't more recent and potentially stronger methods like USM, InstructUIE, and GPT4 considered for benchmarking?

---

> ### Author Response · Authors · 2023-11-17
>
> Before answering your concerns, we would like to emphasize that using code style prompts for IE is not one of our contributions. It was just a development decision due to its practical properties.
>
> Regarding the first weakness, our main contribution is the use of annotation guidelines. The idea for this came from the weaknesses of previous instruction-tuned approaches. We go beyond instruction tuning, as we explore techniques to gear the model towards focusing on the guidelines.
>
> For the second weakness, GoLLIE is not a product. Therefore, it is not intended nor designed to be used by general users. Our goal building GoLLIE was to show that teaching a model how to follow guidelines is beneficial, independently of the prompting strategy choice. In any case, coding skills are not needed to just interpret Python class definitions. Alternatively, it is straightforward to build a user friendly interface on top of it, but this is out of the research scope.
>
> For the third weakness, we will address each point at a time:
> * __Backbone influence:__ Our goal is to demonstrate that training LLMs to adhere to annotation guidelines improves the performance of IE models in unseen tasks.  We demonstrate this by building GoLLIE and a baseline. The baseline shares the same model, same data, same environment but uses no guidelines. InstructUIE is comparable to our baseline, as both receive the same amount of information in the input. The main difference is the backbone LLM , InstructUIE uses FlanT5 instead of LLaMA/CodeLLaMA. Note that [1] indicates that FlanT5 is superior to LLaMA on instruction following tasks, so our backbone starts from a weaker position. As you suggest, we could have added guidelines to InstructUIE. Unfortunately, this is not possible, as Flan-T5 is limited to only 512 tokens in the input. Our research shows that doing this would have improved the performance of InstructUIE in unseen-tasks.  In any case, we do not claim that our choice of backbone and code-style prompting is superior to the framework presented in InstructUIE. Our main claim is that including guidelines in the training and inference of information extraction models improves the performance of the system on unseen tasks.
> * __Prompt style impact:__ Although not rigorously tested, the Code style-syntax worked well in our preliminary experiments with LLaMA (completely zero-shot without any fine-tuning), better than actual natural language prompts. Note that our method involves fine-tuning, and in this case the prompt style loses relevance, as the model rapidly learns to adapt to it (the first checkpoint in the training already showed very few malformed outputs and hallucinations). We expect that Code-style prompts and natural language instructions would perform similarly after fine tuning. Although this would be an interesting experiment,  it would involve an amount of human labor and compute resources that we don't have (implement another system, designing a set of different prompts and running the experiments). Using Code-style syntax is a convenience, not a contribution of our research. Regarding “the necessity of this design for open-source LLMs during instruction tuning remains unproven” , we do not claim that this style is necessary.
>
> For the fourth weakness, under this scenario, the prompt variations are very restricted. All the inputs should follow the template illustrated in Figure 2.  However, we do expect the users to test different class definitions, or guidelines styles. The purpose of the evaluation on un-seen tasks is designed to evaluate how the model performs in this scenario. Each dataset used in the zero-shot evaluation has its own definitions written in its own styles, which allows us to indirectly verify the robustness of our method.
>
> Regarding the last weakness, we plan to explore how the model scales with data in the future. However, the efficiency of the approach is proved, as GoLLIE 7B outperforms 11B Instruct-UIE on zero-shot IE tasks with much less training data: 9 IE datasets on our side vs 1.8k FLAN tasks and 32 IE datasets for Instruct-UIE.
>
> [1] Chia, Y. K., Hong, P., Bing, L., & Poria, S. (2023). INSTRUCTEVAL: Towards Holistic Evaluation of Instruction-Tuned Large Language Models. arXiv preprint arXiv:2306.04757.

---

> > ### Comment · Reviewer_xrh5 · 2023-11-20
> > **Further comments on specific points**
> >
> > Thank you for your detailed response, which has clarified some of my concerns. However, I would like to offer further comments on specific points.
> >
> > 1. **On the Novelty of Using Guidelines**: While the concept of using guidelines as a novel approach in your paper is appreciated, it appears somewhat limited in its novelty. Similar approaches have been explored in past research, notably in a 2019 ACL paper which also enhanced model capabilities through guidelines. This category of work, involving the incorporation of additional knowledge to boost model performance, seems to be under-cited and discussed in your paper.
> > 2. **Backbone Influence**: I mostly agree with the points made in your rebuttal. However, I would like to challenge the assertion that “FlanT5 is superior to LLaMA on instruction following tasks, so our backbone starts from a weaker position.” In the domain of Information Extraction (IE), this conclusion may not be consistently valid unless it is supported by targeted experimental data.
> > 3. **Prompt Style Impact**: I understand the constraints you faced, but if it were possible to supplement your paper with relevant experiments comparing different prompt styles, it would significantly enhance the robustness and credibility of your findings.
> >
> > [1] Description-Based Zero-shot Fine-Grained Entity Typing
> >
> >
> >
> > Reviewer xrh5

---

> > > ### Author Response · Authors · 2023-11-21
> > > **Answer to further comments**
> > >
> > > Thank you for the reference. The way Obeidat et al. (2019) incorporates the label descriptions into the model differs substantially from the current use of LMs. In addition, we believe that our work presents other significant contributions to the field, such as generalization to multiple tasks. We also present an in-depth study of the cases and reasons in which GoLLIE succeeds and fails at following guidelines, which we hope would be of great relevance for future works in the field. We have updated the related work section to add this reference.
> > >
> > > We agree with your second comment.  With our response we wanted to emphasize that our baseline and InstructUIE use the same amount of information as input, but differ on the backbone LLM and train data. The capability comparison between FlanT5 and LLaMA is still an open-research question in the field.
> > >
> > > We will consider  the prompt style impact experiments as an interesting research line for future work.

---

> ### Author Response · Authors · 2023-11-17
> **Regarding the questions**
>
> * __Sampling strategy:__  The only training dataset that has a size that significantly differs from the others is OntoNotes. This dataset is generated semi-automatically and is orders of magnitude larger than the other ones. Therefore, we sampled 30.000 random examples from the training set. The imbalance on dataset size for the rest of the datasets is small. As it can be seen in the supervised evaluation result, our model does not overfit to a single dataset/task. Therefore, we do not consider it necessary to implement any additional sampling strategies.
> * __Scalability with Numerous Labels:__ The inclusion of guidelines in datasets with hundreds of labels will make inputs extremely long, and they would not fit in the context size of current LLMs. This is an already known constraint in LLMs, and recently a lot of research effort is being directed to algorithms that efficiently increase the context window size. We expect that in the future, LLMs will have a context window large enough to fit  not only more labels, but also more detailed guidelines as well. For now, the problem could be solved by batching the labels into multiple input examples. That is, instead of prompting the model with, for example, 100 labels in a single input, it is possible to prompt the model with 10 inputs that incorporate 10 labels each and combine all the outputs into a single one. In any case, we will mention in the paper that this is a limitation for our model. We have added a discussion about this limitation to the updated paper (Appendix E).
> * __QLoRa vs Full-Model finetuning:__ we did provide information about our preliminary analysis in Appendix C.3 (also referred to in section 4.2 “Training details”). We have tested multiple hyperparameter configuration for Full-Model finetuning (we have tested the same hyperparameters used by previous work) and none of it was successful. This doesn’t mean that such a configuration doesn’t exist. What we claim in our paper is that LoRA achieves very good performance and requires much less compute resources than Full-Fine Tuning. Which is supported by previous research and demonstrated in Appendix C.3. We will clarify this in the paper.
> * __Benchmark selection:__ Instruct-UIE is our main baseline (See Section 5.2), which is  the current state-of-the-art Information Extraction model in zero-shot settings. In the Instruct-UIE paper, they already demonstrate that their method is superior to USM, so we didn’t use it in our evaluation (Although we discuss USM in the Related Work Section). Regarding closed-source models such as GPT-4, our objective is to test the hypothesis that training Large Language Models (LLMs) to adhere to annotation guidelines can enhance their zero-shot performance in Information Extraction tasks. Validating this hypothesis involves building a baseline not trained to follow annotation guidelines and performing multiple ablation studies. This is not possible with closed-source models such as GPT-4. Moreover, we know almost nothing from the underlying model(s) powering this commercial product or the data used for training. On top of that, this product is regularly updated, and previous versions are not accessible anymore, therefore, experiments with GPT-4 are not reproducible. Consequently, evaluating GPT-4, or any other closed-source modes, would not add any additional scientific insight in this context of this paper.

---

> > ### Comment · Reviewer_xrh5 · 2023-11-20
> >
> > Thank you for addressing my questions. With these additional response, it looks more solid.
> >
> > Thanks,
> >
> > Reviewer xrh5

---

### Official Review · Reviewer_bJoP · 2023-10-31

**Soundness:** 3 good
**Presentation:** 3 good
**Contribution:** 3 good
**Rating:** 6
**Confidence:** 3

**Summary:**

This paper proposes an LLM fintuned to comply with annotation guidelines for information extraction tasks. Specifically, with a focus on NER, EE, and EAE (19 datasets in total), the task format is unified into a code style: the label class definitions as class docstrings and the candidates as a comment for the arguments.  Some regularization techniques are proposed to ensure that the model follows the guidelines and does not just learn to identify specific datasets. Experiments show that a finetuned version of Code-LLaMA achieves comparable performance compared with a naive baseline finetuned without guidelines, and beats the baseline in zero-shot settings by a large margin.

**Strengths:**

1. This paper proposes to utilize the annotation guidelines to boost the performance of zero-shot information extraction. The idea is intuitive and compelling.
2. The experimental results properly demonstrate the effectiveness of the proposed method and the bonus of the annotation guidelines.
3. The presentation of this paper is clear, with few typos.

**Weaknesses:**

1. The motivation behind the code style and its effects are unclear. Note that incorporating the annotation guidelines in a natural language style rather than the code style is also feasible. The LLMs pretrained on codes may lack natural language understanding capabilities that are crucial to conducting IE tasks. This may result in the low performance of the baseline. The advantages may lie in the capability of generating outputs following code grammar, which mitigates the need for parsing outputs. The authors should justify the necessity of the code style and its underlying effects compared to the natural language style.
2. The ablation studies of class order shuffling and guideline paraphrasing are missing.

**Questions:**

1. How many samples are used to supervise finetuning LLaMA?
2. The guidelines used in this paper are specifically the basic definitions of entity/relation/event types, which should have been provided to LLMs. However, the full annotation guidelines usually cover many edge cases and may contain tens of pages. Have the authors considered incorporating more fine-grained guidelines?

---

> ### Author Response · Authors · 2023-11-17
>
> The use of code (more precisely Python syntax) as an underlying format for the input and output was primarily motivated by previous research works [1, 2]. Prompting IE tasks requires some kind of structured format for the input and output. As you mention, we could also have prompted the model with more natural style language. However, we decided to use a format that is familiar for both the human and the model, in order to represent the input-output. Previous work has demonstrated that LLMs trained on code are capable of performing  IE [1, 2]. Furthermore, most LLMs fine-tuned for code generation, including Codellama, are fine-tuned on top of a general purpose LLM and retain most of its natural language understanding capabilities. Additionally, our method outperforms Instruct-UIE [3] on zero-shot, a model prompted with natural language.
>
> That being said, using code-style format is not strictly our contribution; so we did not dig deeper on the matter. It is more of a convenience for easily parsing the input and output, than a decision that contributes to the model performance. We believe that, since fine-tuning is involved, LLMs will easily adapt to any input structure, that being code-style format, natural language instructions or any other alternative. While performing this evaluation would have great interest, we currently lack the human and computer resources for such experiments.
>
> Regarding the “This may result in the low performance of the baseline.”, both the baseline and GoLLIE share the same conditions, except for the mentioned contributions.
>
> With respect to the lack of ablation of guideline shuffling and paraphrasing, we have updated the paper to include both.
>
> Regarding your question about how many samples we have used to train the model, the models were trained for around 15k steps (3 epochs of our entire dataset with a batch size of 32). Note that for a single dataset a single sample might appear several times as we generate different inputs per task. We have updated the paper to include this information in Appendix D.2.
>
> Regarding your question about the full annotation guidelines, it is something that we definitely want to try. However those guidelines are very long, and considering the amount of labels we have per example, we found them intractable with our current models/infrastructure setup. With the recent efficient methods and models for larger sequences we believe we could soon explore the use of more detailed and longer guidelines in our experiments. However, it is important to note that in the paper we use some of the more basic guidelines to illustrate our method, since we consider that they were the easiest to understand. We have included a new figure (Figure 8, in Appendix E) that shows the guideline for the entity Point in the HarveyNer dataset. This guideline is very descriptive including edge cases and exceptions. All the guidelines we use will be made publicly available and we will add a link to them in the paper.
>
>
> [1] Xingyao Wang, Sha Li, and Heng Ji. 2023. Code4Struct: Code Generation for Few-Shot Event Structure Prediction. In Proceedings of the 61st Annual Meeting of the Association for Computational Linguistics (Volume 1: Long Papers), pages 3640–3663, Toronto, Canada. Association for Computational Linguistics.
>
> [2] Peng Li, Tianxiang Sun, Qiong Tang, Hang Yan, Yuanbin Wu, Xuanjing Huang, and Xipeng Qiu. 2023. CodeIE: Large Code Generation Models are Better Few-Shot Information Extractors. In Proceedings of the 61st Annual Meeting of the Association for Computational Linguistics (Volume 1: Long Papers), pages 15339–15353, Toronto, Canada. Association for Computational Linguistics.
>
> [1] Wang, X., Zhou, W., Zu, C., Xia, H., Chen, T., Zhang, Y., ... & Du, C. (2023). InstructUIE: Multi-task Instruction Tuning for Unified Information Extraction. arXiv preprint arXiv:2304.08085.

---

> ### Author Response · Authors · 2023-11-21
> **Ablation updated**
>
> In the last update of the paper we have added the missing ablation studies. As expected, the shuffle and paraphrase did not have a significant effect.

---

> > ### Comment · Reviewer_bJoP · 2023-11-23
> >
> > Thank you for your reply.

---

> ### Public Comment · ~Claude_Ross1 · 2024-07-31
> **Thanks**
>
> Thank you for your feedback. If you are looking for the best essay writing service because you are unable to complete your college assignments, the best post for you is at https://basketballstars-game.io. The essay writing samples on this website are free to use, and you can use them to write an essay on any topic. You can also share this page with your friends.

---

### Official Review · Reviewer_y8gw · 2023-11-01

**Soundness:** 4 excellent
**Presentation:** 3 good
**Contribution:** 3 good
**Rating:** 6
**Confidence:** 4

**Summary:**

This paper focuses on zero-shot information extraction. Their goal is to train a large model that can generalize well to unseen domains and datasets. They main idea is using code as the intermediate layer for all the information extraction tasks and considering annotation guidelines or additional information during pre-training or testing. This additional information makes the model to understand tasks better and therefore be able to generalize. Experiments demonstrate the potential of the proposed method.

**Strengths:**

- The performance is impressive.
- The pre-trained model is valuable.

**Weaknesses:**

- My main concern is as follows. Although the authors try to split the datasets for training and testing based on domains. I believe that there is still a large overlap among their output spaces. For example, RAMS, which is considered as one of the training tasks, contains role label *transporter*, *vehicle*, and *place*, while WikiEvents, which is consider as one of the testing tasks, also contains those role labels. In NER tasks, I believe the same situation happens as well. This makes the setting not *truly* zero-shot because the model has already learned those concepts. The model can get improvements just because including more *in-domain* supervised training signals.
- I think the authors should be careful when referring the state-of-the-art models. In fact, in Table 2, the reported SOTA numbers and models are not SOTA anymore.

**Questions:**

- See above.

---

> ### Author Response · Authors · 2023-11-17
>
> About the zero-shot evaluation: In order to compare GoLLIE with previous work, we decided to use as zero-shot datasets the same datasets that have been used for this purpose in previous works, such as a InstructUIE [1], USM [2] or UniNER [3] among others. These methods use an even larger set of pre-training datasets for their models, so more entities in the zero-shot datasets overlap with the ones in the training dataset. Thus, even with less overlap,, GoLLIE outperforms all of them. It is also noteworthy that, while some datasets share similar entities, the annotation guidelines often vary between them. For example, ACE05  annotates pronouns as persons, while, CoNLL03 does not. In any case, we agree with your comment: as we also commented in the paper (first paragraph of Section 5.2), such overlap could make both the baseline and CoLLIE perform better in the evaluation. We believe that understanding the model performance with totally unseen labels is very important. Therefore, we have updated the paper to include an analysis regarding seen and unseen label evaluation in Section 5.2. This analysis is further extended in the Appendix B. As a short summary, the baseline model achieves significantly lower F1 scores on unseen labels than on seen labels. In contrast, GoLLIE demonstrates a more robust performance, showing a smaller gap in F1 scores between the seen and unseen labels.
>
> Regarding the SoTA models in Table 2 (supervised results), the main purpose of this paper is not to improve the supervised SoTA, but to focus on the zero-shot generalization. Our results on Table 2 are shown for completeness and to show that the use of guidelines does not hurt performance when compared to the baseline. The SoTa results reported there correspond to systems that share the most comparable settings to us. In order to be clear we have updated the text to make it more explicit. The final version will include results for the best SoTA models.
>
> [1] Wang, X., Zhou, W., Zu, C., Xia, H., Chen, T., Zhang, Y., ... & Du, C. (2023). InstructUIE: Multi-task Instruction Tuning for Unified Information Extraction. arXiv preprint arXiv:2304.08085.
>
> [2] Lou, J., Lu, Y., Dai, D., Jia, W., Lin, H., Han, X., ... & Wu, H. (2023). Universal Information Extraction as Unified Semantic Matching. arXiv preprint arXiv:2301.03282.
>
> [3] Zhou, W., Zhang, S., Gu, Y., Chen, M., & Poon, H. (2023). Universalner: Targeted distillation from large language models for open named entity recognition. arXiv preprint arXiv:2308.03279.

---

> > ### Comment · Reviewer_y8gw · 2023-11-19
> >
> > Thanks for your response.

---

### Author Response · Authors · 2023-11-17
**Overall response to reviewers**

We thank the reviewers for their thorough reviews which we believe help to improve the paper. We have updated our paper according to the reviewers comments.

Changes on the new version of the paper:
* Add clarifications about CASIE evaluation. (Section 4.1)
* Update results of 13B and 34B models with another run to match the total runs (3) of the rest (Tables 2 and 3). This has resulted in a minor variation of the f1-scores for the 13B and 34B models.
* Add references for multiple Appendix sections in the main text of the paper so the reader is aware of the further details/experiments in the Appendix.
* [suggested by @zWZU] Add missing references. (Section 2)
* [suggested by @xrh5 and @zWZU] Add discussion about the implications of prompt length, scalability and code format overhead to the Appendix. (Referenced in Section 3.1)
* [suggested by @zWZU] Add to the Appendix a discussion of the human effort required to build and format the datasets. (Referenced in Section 4.1)
* [suggested by @y8gw] Updated references to the SoTA. (Section 4.2)
* [suggested by @y8gw] Add a Seen/Unseen label analysis to the Section 5.2 with an extended analysis in the Appendix B (Referenced in Section 5.2 and 6)
* [suggested by @bJoP] Add “paraphrases” ablation results. (Section 5.4)
* [suggested by @bJoP] Add “shuffle” ablation results. (Section 5.4)
* [suggested by @zWZU] Update Table 6 to be more user-friendly.
Add an example of generalization to a new custom task in Appendix A.1.
* [suggested by @bJoP] Include details about the number of examples for each dataset and total samples / steps used for training (Appendix D.2)

---

### Meta-Review · Area_Chair_Lx5T · 2023-12-10

**Metareview:**

The paper proposes a novel approach to enhance zero-shot information extraction (IE) by introducing GoLLIE, a Guideline-following Large Language Model for IE. The model is fine-tuned to comply with annotation guidelines represented in a Python code-based format. Comprehensive evaluation demonstrates that GoLLIE is able to generalize to and follow unseen guidelines, outperforming previous attempts at zero-shot information extraction.

Strengths: The paper's main strength lies in its innovative approach to leveraging annotation guidelines in the context of zero-shot information extraction. Reviewers acknowledge the impressive performance and value of the pre-trained model, commends the intuitive and compelling idea, with experimental results demonstrating the effectiveness of the proposed method. The paper is well-written, with clear presentation and minimal typos. The comprehensive error analysis provided in the paper offers valuable insights into the challenges of zero-shot IE.

Weaknesses: The weaknesses highlighted by the reviewers center around several key aspects.
1. Concerns about the overlap in the output spaces of training and testing datasets, potentially limiting the true zero-shot nature of the model.
2. The motivation behind the code style representation and calls for justification of its necessity compared to a natural language style.
3. The claim of using "annotation guidelines" might be overstated, as the paper mainly considers label names, descriptions, and few-shot examples, potentially missing the complexity captured by true IE annotation guidelines.

**Justification For Why Not Higher Score:**

While the paper introduces an intriguing approach and demonstrates promising results in zero-shot IE tasks, the weaknesses highlighted by the reviewers need to be carefully addressed. Specifically, the overlap in output spaces, justification for the code style representation, and the need for more comprehensive experimental validation are crucial considerations.

**Justification For Why Not Lower Score:**

Reviewers are generally positive and the weaknesses don't warrant a necessary rejection.

---

### Decision · Program_Chairs · 2024-01-16

Accept (poster)